# Evaluation of a Hybrid Underwater Sound-Absorbing Metastructure by Using the Transfer Matrix Method

**DOI:** 10.3390/ma16041718

**Published:** 2023-02-18

**Authors:** Han-Chun Lin, Shu-Cheng Lu, Hsin-Haou Huang

**Affiliations:** Department of Engineering Science and Ocean Engineering, National Taiwan University, Taipei 10617, Taiwan

**Keywords:** sound-absorbing metastructure, microperfored panel, viscoelastic substrate, resonance, finite-element method, transfer matrix method, integrated transfer matrix (ITM) method

## Abstract

In this study, we designed a novel hybrid underwater sound-absorbing material of the metastructure that contains a viscoelastic substrate with a microperforated panel. Two types of sound-absorbing metastructures were combined to achieve satisfactory sound absorption performance in the low-frequency range. A homogenized equivalent layer and the integrated transfer matrix method were used to theoretically evaluate the sound absorption performance of the designed nonhomogeneous hybrid metastructure. The theoretical results were then compared with the results obtained using the finite-element method. The designed hybrid sound-absorbing metastructure exhibited two absorption peaks because of its different sound-absorbing mechanisms. The acoustic performance of the developed metastructure is considerably better than that of a traditional sound absorber, and the sound absorption coefficient of the developed metastructure is 0.8 in the frequency range of 3–10 kHz. In addition, an adjustment method for the practical underwater application of the designed metastructure is described in this research. Further studies show that the sound absorption coefficient of the adjusted metastructure still has 0.75 in the frequency range of 3–10 kHz, which indicates that this metastructure has the potential to be used as an underwater sound-absorbing structure. The results of this study can be used as a reference in the design of other novel hybrid underwater sound-absorbing structures.

## 1. Introduction

Many studies have attempted to design effective structures and materials to achieve noise suppression. Viscoelastic materials such as rubber and polyurethane absorb sound waves in sea water when they exhibit good impedance matching with sea water. Further, friction is generated in these materials when sound waves pass through them, which results in them absorbing sound waves [1,2]. However, developing a broadband underwater sound absorber is difficult. Toward the end of World War II, the Germans developed a synthetic rubber structure with a cavity and installed it on submarines for acoustic stealth. To analyze this structure, Jackins and Gaunaurd used resonance scattering theory to predict the reflection of sound waves from a bilaminar structure composed of rubber and a spherical air-filled perforation [3]. Ivansson presented a layer-multiple-scattering method to investigate thin rubber coatings with cavities in a doubly periodic lattice [4]. Tao presented a simplified and computationally efficient analytical model to solve the characteristic equation and determine the wavenumber of axisymmetric waves. The sound absorption coefficient of an underwater structure with periodic cavities can be evaluated rapidly by using the transfer matrix method [5]. Over time, the shapes of sound absorbers have gradually become increasingly complicated with multiple connected parts and complex acoustic elements. Vijayasree and Munjal developed an analytical approach for predicting the sound-absorbing ability of complex absorbers [6]. Subsequently, many studies have used the transfer matrix method to optimize the shape of the cavity for low-frequency sound absorption [5,7,8]. To broaden the sound absorption bandwidth, various absorption structures have been designed, such as phononic glass structures [9], multilayered locally resonant scatterers [10], metastructures exhibiting the coupling effect between torsion vibration and longitudinal wave deformation [11], and piezoelectric composites [12,13,14].

In 1997, a microperforated panel (MPP) was developed to increase the bandwidth of sound absorption in the relatively low-frequency region. The bandwidth of this plate was enhanced by reducing its aperture size and matching its flow resistance with the characteristic impedance of the surrounding medium [15]. Lee and Kwon employed the transfer matrix method to estimate the absorption performance of multilayer perforated panel systems [16]. Tang et al. combined a perforated structure with a honeycomb–corrugated composite core. This hybrid structure exhibited broadband low-frequency sound absorption and excellent mechanical strength [17]. Because of the high hydrostatic pressure and complicated conditions in the sea, limited research has been conducted on underwater sound-absorbing materials. Wang was the first to indicate that the feasibility of using perforated plates as underwater sound absorbers depends on panel thickness and pore size [18]. Li et al. investigated a new type of sound-absorbing structure containing an MPP and a viscoelastic substrate and determined that this structure achieved broadband sound absorption [19].

In the present study, we developed a hybrid sound-absorbing structure that contains a viscoelastic substrate with an MPP. The MPP can absorb low-frequency sound energy, whereas the viscoelastic substrate, which contains cavities, absorbs high-frequency sound energy; thus, the developed hybrid structure achieves a wide underwater sound absorption bandwidth. In this research, first, we calculated the sound absorption coefficient of the developed structure by using the transfer matrix method. Second, we used the finite-element software COMSOL Multiphysics to simulate the sound absorption metastructure. The finite-element simulation results confirmed the applicability of the transfer matrix method for evaluating the sound absorption coefficient of a metastructure and revealed the absorption mechanism of the developed hybrid sound absorption metastructure. Finally, the method for using the developed hybrid metastructure is described in this paper.

## 2. Theoretical Formulation

### 2.1. Theoretical Model

Figure 1a shows a schematic of the developed hybrid sound-absorbing metastructure containing periodically arranged cavities. As displayed in Figure 1b, each unit of the developed hybrid metastructure comprises an elastomer and MPP. The MPP is composed of a highly rigid material with cavities filled with water. The elastomer is viscoelastic with cavities filled with air. The steel backing of this metastructure is considered a rigid wall; thus, all sound waves are assumed to be reflected back from the bottom of the metastructure. Because the overall structure includes cylindrical cavities with discontinuous cross-sectional areas and MPPs, the integrated transfer matrix (ITM) method was used to analyze the structure [6]. In this method, the overall metastructure is divided into numerous sections, and a transfer matrix that relates the variables of the incidence and terminal surfaces is generated for each section. Finally, the transfer matrix of each section is combined with the boundary conditions between the sections to obtain the final transfer matrix.

Figure 2a displays a schematic of an approximate model of the developed metastructure. The theoretical analysis of hexagonal elements is difficult; therefore, the hexagonal elements were approximated into cylindrical elements of equal volume. In the ITM method, the overall metastructure is divided into several acoustic elements, which are indicated by red numbers in Figure 2b. Acoustic element 1 is composed of an MPP and a water-filled cavity. Acoustic elements 2, 3, and 5 are cylindrical elastomers with different cross-sectional areas. Acoustic element 4 is an elastomer layer with an air-filled cavity. The black numbers in Figure 2b represent the intersection of different sections. The two-dimensional plot of the cut plane B–B of the analysis model is displayed in Figure 2b.

The ITM method can be used to evaluate the sound absorption coefficient of an overall structure by combining the transfer matrices of all sections into a single matrix. The sound pressure and volume velocity are used to express the complexity of the wave propagating into the material. When a sound wave perpendicularly impinges on an acoustic metastructure, the two surfaces of the acoustic metastructure can be described by a linear relationship. Therefore, the sound pressure and volume velocity of the incidence surface and terminal surface can be expressed as follows:(1)$$\left[\begin{array}{c} p_{i}\\ U_{i} \end{array}\right]=\left[\begin{array}{cc} M_{11} & M_{12}\\ M_{21} & M_{22} \end{array}\right]\left[\begin{array}{c} p_{t}\\ U_{t} \end{array}\right],$$
where $p_{i}$ and $p_{t}$ are the sound pressures at the incidence and terminal surfaces, respectively, and $U_{i}$ and $U_{t}$ are the volume velocities at the incidence and terminal surfaces, respectively.

### 2.2. Theoretical Calculations

#### 2.2.1. Transfer Matrix for a Uniform Tube

In the acoustic wave equation, the medium is assumed to be stationary. Thus, the sound pressure and volume velocity on both sides of a uniform pipe can be related using the following matrix [20]:(2)$${\left[P\right]}_{2\times 2}=\left[\begin{array}{cc} \cos\left(kL\right) & iZ\sin\left(kL\right)\\ \frac{i}{Z}\sin\left(kL\right) & \cos\left(kL\right) \end{array}\right],$$
where k=ω/c, $Z=\frac{\rho c}{S},\;S$, and L are the wave number, characteristic acoustic impedance, cross-sectional area of the pipe, and pipe length, respectively.

#### 2.2.2. Transfer Matrix Method for an MPP

Figure 3 displays an MPP that can be regarded as a parallel arrangement of many thin-plate units with circular openings. One unit is marked by a red dashed box in the figure. Because of the small diameter of the cylindrical openings of the MPP, the influence of the viscous boundary layer must be considered.

The pressure difference between the two ends of the tube is expressed as follows [15]:(3)$$\rho \frac{\partial v}{\partial t}-\frac{\eta }{r}\frac{\partial }{\partial r}\left(r\frac{\partial v}{\partial r}\right)=\frac{\Delta P}{t},$$
where ρ is the density of the fluid in the tube, η is the dynamic viscosity coefficient of the fluid, v is the fluid particle velocity, and t is the thickness of the panel. Assuming the existence of harmonic motion, the time differential term of Equation (3) is replaced by iω. On the basis of the equation of motion in the tube, the specific acoustic impedance is expressed as follows:(4)$$Z_{h}=\frac{\Delta P}{\bar{v}}=i\omega \rho t{\left[1-\frac{2}{k_{c}\sqrt{-i}}\frac{J_{1}\left(k_{c}\sqrt{-i}\right)}{J_{0}\left(k_{c}\sqrt{-i}\right)}\right]}^{-1},$$
where $k_{c}=d\sqrt{\rho \omega /4\eta }$ is the perforation constant, which is proportional to the diameter of the tube and the thickness of the viscous boundary layer, $\bar{v}$ is the average particle velocity of the cross section of the tube, and d is the hole diameter of the MPP. The end corrections caused by the friction between the flowing medium and the surface of the plate and between the flowing medium and sound waves must be added to Equation (4). Assuming that no resonance interaction occurs between the cylindrical cavities, the specific acoustic impedance of the MPP can be obtained by dividing the specific acoustic impedance of a single tube by the porosity ϕ as follows:(5)$$Z_{m}=\frac{i\omega \rho t}{\phi }{\left[1-\frac{2}{k_{c}\sqrt{-i}}\frac{J_{1}\left(k_{c}\sqrt{-i}\right)}{J_{0}\left(k_{c}\sqrt{-i}\right)}\right]}^{-1}+\frac{\sqrt{2}\mu k_{c}}{\phi d}+i\frac{0.85\omega \rho d}{\phi }.$$

Assuming that the thickness t is low, the volume velocity on both surfaces of the MPP is the same; thus, the acoustic variables on both sides of the MPP can be expressed using a 2 × 2 transfer matrix as follows:(6)$${\left[M\right]}_{2\times 2}=\left[\begin{array}{cc} 1 & Z_{m}/S\\ 0 & 1 \end{array}\right].$$

#### 2.2.3. Transfer Matrix Method for an Underwater Metastructure with an Air-Filled Cavity

In this section, an underwater metastructure with an air-filled cavity is analyzed using the simplified theoretical model developed in [5,7]. Figure 4 displays a cross-sectional view of a section of a sound-absorbing metastructure with a cylindrical cavity. The polar coordinate system is used, and the red dashed line represents the central axis of the cylindrical model. Assuming that a harmonic wave is incident on a cylindrical elastic material, the motion equation can be written as follows [21]:(7)$$\mu \nabla^{2}u+\left(\lambda +\mu \right)\nabla \left(\nabla \cdot u\right)=-\rho_{e}\omega^{2}u,$$
where u is the displacement vector, λ and μ are the Lamé constants, $\rho_{e}$ is the density of the viscoelastic material, and ∇ and $\nabla^{2}$ are the gradient operator and Laplacian operator, respectively. The displacement vector generated by wave propagation in a solid can be expressed as follows:(8)u=∇ϕ+∇×Ψ,
where ϕ is a function of the expansion wave part and Ψ is a function of the distortion wave part expressed as follows:(9)$$\left(\nabla^{2}+k_{l}^{2}\right)\phi =0,$$
(10)$$\left(\nabla^{2}+k_{t}^{2}\right)\psi =0,$$
where kl=ωcl is the longitudinal wavenumber, kt=ωct is the transverse wave number.$\;c_{l}$ and $c_{t}$ are the longitudinal and transverse wave speeds, respectively. The axisymmetric relationship function of a cylinder ϕ and the function Ψ can be simplified and written as follows:(11)$$\phi =\left[AJ_{0}\left(k_{l,r}r\right)+BY_{0}\left(k_{l,r}r\right)\right]e^{ik_{z}z},$$
(12)$$\psi =\left[CJ_{0}\left(k_{t,r}r\right)+DY_{0}\left(k_{t,r}r\right)\right]e^{ik_{z}z},$$
where A, B, C, and D are the undetermined coefficients, $k_{l,r}$ and $k_{t,r}$ are the radial wavenumbers of longitudinal and transverse waves, $k_{z}$ is the axial wavenumber, $J_{0}$ and $Y_{0}$ are the Bessel functions of the first and second kind of the zero order. The correlation of the above wavenumbers can be written as follows:(13)$$k_{l}^{2}=k_{l,r}^{2}+k_{z}^{2},$$
(14)$$k_{t}^{2}=k_{t,r}^{2}+k_{z}^{2}.$$

In cylindrical coordinate system, the radial displacement $u_{r}$ and axial displacement $u_{z}$ can be expressed as follows:(15)$$u_{r}=\frac{\partial \phi }{\partial r}+\frac{\partial^{2}\psi }{\partial r\partial z},$$
(16)$$u_{z}=\frac{\partial \phi }{\partial z}-\frac{\partial^{2}\psi }{\partial r^{2}}-\frac{\partial \psi }{r\partial r}.$$

The strain can be expressed as follows:(17)$$\varepsilon_{r}=\frac{\partial u_{r}}{\partial r},$$
(18)$$\varepsilon_{\theta }=\frac{u_{r}}{r},$$
(19)$$\varepsilon_{z}=\frac{\partial u_{z}}{\partial z},$$
(20)$$\varepsilon_{rz}=\frac{\partial u_{z}}{\partial r}+\frac{\partial u_{r}}{\partial z}.$$

By combining the aforementioned equation of motion with the expressions of the elastic material’s stress, Equations (21) and (22), the displacement and stress variables of the aforementioned section can be expressed in matrix form as Equation (23):(21)$$\sigma_{r}=\lambda \left(\varepsilon_{r}+\varepsilon_{\theta }+\varepsilon_{z}\right)+2\mu \varepsilon_{r},$$
(22)$$\tau_{rz}=\mu \varepsilon_{rz},$$
(23)$$\left\{\begin{array}{c} u_{r}\\ \sigma_{r}\\ \tau_{rz} \end{array}\right\}=e^{ik_{z}z}\left[\begin{array}{cccc} M_{1,1} & M_{1,2} & M_{1,3} & M_{1,4}\\ M_{2,1} & M_{2,2} & M_{2,3} & M_{2,4}\\ M_{3,1} & M_{3,2} & M_{3,3} & M_{3,4} \end{array}\right]\left\{\begin{array}{c} A\\ B\\ C\\ D \end{array}\right\}.$$

The expressions of the 12 parameters in the aforementioned coefficient matrix (Equation (23)) are presented in Appendix A. Because of the periodic arrangement of the cavities of the developed sound-absorbing metastructure, the interface between adjacent structural units should satisfy the following balance conditions. First, the radial displacement and tangential stress at $r=r_{A}$ must be equal to 0. Second, considering the impedance difference between air and the viscoelastic material, no radial stress and tangential stress should exist at $r=r_{c}$. By substituting the aforementioned boundary conditions into Equation (23), the coefficients *A*, *B*, *C*, and *D* can be determined as follows:(24)$$\left[\begin{array}{cccc} M_{1,1}(r_{A}) & M_{1,2}(r_{A}) & M_{1,3}(r_{A}) & M_{1,4}(r_{A})\\ M_{2,1}(r_{c}) & M_{2,2}(r_{c}) & M_{2,3}(r_{c}) & M_{2,4}(r_{c})\\ M_{3,1}(r_{c}) & M_{3,2}(r_{c}) & M_{3,3}(r_{c}) & M_{3,4}(r_{c})\\ M_{3,1}(r_{A}) & M_{3,2}(r_{A}) & M_{3,3}(r_{A}) & M_{3,4}(r_{A}) \end{array}\right]\{\begin{array}{c} A\\ B\\ C\\ D \end{array}\}=0.$$

When Equation (24) has a nonzero solution, the determinant of the coefficient matrix in this equation is 0. By solving the roots of the determinant, a multiorder wave number $k_{n}$ (*n* = 1,2, 3…) can be obtained, and the sound absorption efficiency of the developed metastructure at low frequencies mainly depends on the first-order axisymmetric wave number. The higher-order wave number is ignored, and only the first wave number $k_{1}$ is considered. On the basis of the aforementioned calculation, the effective wavenumber $\bar{k}$, effective density $\bar{\rho },$ and effective characteristic acoustic impedance $\bar{Z}$ of the examined section can be expressed as follows:(25)$$\bar{k}=k_{1},$$
(26)$$\bar{\rho }=\left[1-{\left(\frac{r_{c}}{r_{A}}\right)}^{2}\right]\rho_{e}+{\left(\frac{r_{c}}{r_{A}}\right)}^{2}\rho_{a},$$
(27)$$\bar{Z}=\bar{\rho }\bar{c},$$
where the corresponding effective sound velocity $\bar{c}=\omega /\bar{k}$ and $\rho_{a}$ is the air density. The simple analysis model can homogenize the structure of the examined section, and the 2 × 2 transfer matrix of this section is expressed as follows:(28)$${\left[B\right]}_{2\times 2}=\left[\begin{array}{cc} \cos\left(\bar{k}L_{c}\right) & i\bar{Z}\sin\left(\bar{k}L_{c}\right)\\ \frac{i\sin\left(\bar{k}L_{c}\right)}{\bar{Z}} & \cos\left(\bar{k}L_{c}\right) \end{array}\right].$$

### 2.3. Absorption Coefficient

After calculating the transfer matrices of the different sections of the examined sound-absorbing metastructure, an ITM can be produced to calculate the sound absorption coefficient of the overall structure. Let $T_{i,j\_k}$ be the 2 × 2 transfer matrix of the *i*th acoustic element in section (*j_k*). Section (0_1) in Figure 2b is composed of acoustic elements 1 and 2, for which the individual transfer matrices are known. The transfer matrix in section (0_1) can be expressed as follows:(29)$$\left[\begin{array}{c} p_{1,0}\\ U_{1,0}\\ p_{2,0}\\ U_{2,0} \end{array}\right]=\left[\begin{array}{cc} {\left[T_{1,0\_1}\right]}_{2\times 2} & {\left[0\right]}_{2\times 2}\\ {\left[0\right]}_{2\times 2} & {\left[T_{2,0\_1}\right]}_{2\times 2} \end{array}\right]\left[\begin{array}{c} p_{1,1}\\ U_{1,1}\\ p_{2,1}\\ U_{2,1} \end{array}\right],$$
where [0] is the zero matrix. The transfer matrices of acoustic elements 1 and 2 can be written as follows:(30)$$T_{1,0\_1}=\left[\begin{array}{cc} 1 & \frac{Z_{m}}{S_{1,0_{1}}}\\ 0 & 1 \end{array}\right]\left[\begin{array}{c} \begin{array}{cc} \cos\left(k_{1,0_{1}}L_{1,0_{1}}\right) & iZ_{1,0_{1}}\sin\left(k_{1,0_{1}}L_{1,0_{1}}\right)\\ \frac{i}{Z_{1,0_{1}}}\sin\left(k_{1,0_{1}}L_{1,0_{1}}\right) & \cos\left(k_{1,0_{1}}L_{1,0_{1}}\right) \end{array} \end{array}\right],$$
(31)$$T_{2,0\_1}=\left[\begin{array}{cc} \cos\left(k_{2,0\_1}L_{2,0\_1}\right) & iZ_{2,0\_1}\sin\left(k_{2,0\_1}L_{2,0\_1}\right)\\ \frac{i}{Z_{2,0\_1}}\sin\left(k_{2,0\_1}L_{2,0\_1}\right) & \cos\left(k_{2,0\_1}L_{2,0\_1}\right) \end{array}\right],$$
where $k_{i,j\_k}=\omega /c_{i,j\_k}$, $Z_{i,j\_k}=\rho_{i,j\_k}c_{i,j\_k}/S_{i,j\_k}$, $S_{i,j\_k}$, and $L_{i,j\_k}$ are the wave number, characteristic acoustic impedance, cross-sectional area of the pipe, and pipe length of the *i*th acoustic element between the section (*j_k*), respectively. The transfer matrix of section (1_2) can be written as follows:(32)$$\left[\begin{array}{c} p_{3,1}\\ U_{3,1} \end{array}\right]={\left[T_{3,1\_2}\right]}_{2\times 2}\left[\begin{array}{c} p_{3,2}\\ U_{3,2} \end{array}\right].$$

By inversing the matrices presented in Equations (29) and (32), the following equations are obtained:(33)$$\left[\begin{array}{c} p_{1,1}\\ U_{1,1}\\ p_{2,1}\\ U_{2,1} \end{array}\right]=\left[H\right]\left[\begin{array}{c} p_{1,0}\\ U_{1,0}\\ p_{2,0}\\ U_{2,0} \end{array}\right],$$
(34)$$\left[\begin{array}{c} p_{3,2}\\ U_{3,2} \end{array}\right]=\left[F\right]\left[\begin{array}{c} p_{3,1}\\ U_{3,1} \end{array}\right].$$

For sections with discontinuous areas, the pressure and sound mass velocity at the interface must be equal.
(35)$$p_{1,0}=p_{2,0},$$
(36)$$p_{1,1}=p_{2,1}=p_{3,1},$$
(37)$$\rho_{1,0\_1}U_{1,1}+\rho_{2,0\_1}U_{2,1}=\rho_{3,1\_2}U_{3,1},$$
where $\rho_{i,j\_k}$ is the density of the medium in the *i*th acoustic element in section (j_k). Equations (33) and (34) and the boundary conditions can be rearranged into a single matrix, and this matrix can be rewritten using the Gaussian elimination method to express the acoustic variable relationship in section (0_2) as follows:(38)$$\left[\begin{array}{cccccccccc} H_{1,1} & H_{1,2} & H_{1,3} & H_{1,4} & -1 & 0 & 0 & 0 & 0 & 0\\ H_{2,1} & H_{2,2} & H_{2,3} & H_{2,4} & 0 & -1 & 0 & 0 & 0 & 0\\ H_{3,1} & H_{3,2} & H_{3,3} & H_{3,4} & 0 & 0 & -1 & 0 & 0 & 0\\ H_{4,1} & H_{4,2} & H_{4,3} & H_{4,4} & 0 & 0 & 0 & -1 & 0 & 0\\ 1 & 0 & -1 & 0 & 0 & 0 & 0 & 0 & 0 & 0\\ 0 & 0 & 0 & 0 & 1 & 0 & 0 & 0 & -1 & 0\\ 0 & 0 & 0 & 0 & 0 & 0 & 1 & 0 & -1 & 0\\ 0 & 0 & 0 & 0 & 0 & \rho_{1,0\_1} & 0 & \rho_{2,0\_1} & 0 & -\rho_{3,1\_2}\\ 0 & 0 & 0 & 0 & 0 & 0 & 0 & 0 & F_{1,1} & F_{1,2}\\ 0 & 0 & 0 & 0 & 0 & 0 & 0 & 0 & F_{2,1} & F_{2,2} \end{array}\right]\left[\begin{array}{c} p_{1,0}\\ U_{1,0}\\ p_{2,0}\\ U_{2,0}\\ p_{1,1}\\ U_{1,1}\\ p_{2,1}\\ U_{2,1}\\ p_{3,1}\\ U_{3,1} \end{array}\right]=\left[\begin{array}{c} 0\\ 0\\ 0\\ 0\\ 0\\ 0\\ 0\\ 0\\ p_{3,2}\\ U_{3,2} \end{array}\right].$$

The inverse matrix of the matrix presented in Equation (38) is expressed as follows:(39)$$\left[\begin{array}{c} p_{1,0}\\ U_{1,0}\\ p_{2,0}\\ U_{2,0}\\ p_{1,1}\\ U_{1,1}\\ p_{2,1}\\ U_{2,1}\\ p_{3,1}\\ U_{3,1} \end{array}\right]=\left[\begin{array}{cccccccccc} R_{1,1} & R_{1,2} & R_{1,3} & R_{1,4} & R_{1,5} & R_{1,6} & R_{1,7} & R_{1,8} & R_{1,9} & R_{1,10}\\ R_{2,1} & R_{2,2} & R_{2,3} & R_{2,4} & R_{2,5} & R_{2,6} & R_{2,7} & R_{2,8} & R_{2,9} & R_{2,10}\\ R_{3,1} & R_{3,2} & R_{3,3} & R_{3,4} & R_{3,5} & R_{3,6} & R_{3,7} & R_{3,8} & R_{3,9} & R_{3,10}\\ R_{4,1} & R_{4,2} & R_{4,3} & R_{4,4} & R_{4,5} & R_{4,6} & R_{4,7} & R_{4,8} & R_{4,9} & R_{4,10}\\ R_{5,1} & R_{5,2} & R_{5,3} & R_{5,4} & R_{5,5} & R_{5,6} & R_{5,7} & R_{5,8} & R_{5,9} & R_{5,10}\\ R_{6,1} & R_{6,2} & R_{6,3} & R_{6,4} & R_{6,5} & R_{6,6} & R_{6,7} & R_{6,8} & R_{6,9} & R_{6,10}\\ R_{7,1} & R_{7,2} & R_{7,3} & R_{7,4} & R_{7,5} & R_{7,6} & R_{7,7} & R_{7,8} & R_{7,9} & R_{7,10}\\ R_{8,1} & R_{8,2} & R_{8,3} & R_{8,4} & R_{8,5} & R_{8,6} & R_{8,7} & R_{8,8} & R_{8,9} & R_{8,10}\\ R_{9,1} & R_{9,2} & R_{9,3} & R_{9,4} & R_{9,5} & R_{9,6} & R_{9,7} & R_{9,8} & R_{7,9} & R_{9,10}\\ R_{10,1} & R_{10,2} & R_{10,3} & R_{10,4} & R_{10,5} & R_{10,6} & R_{10,7} & R_{10,8} & R_{10,9} & R_{10,10} \end{array}\right]\left[\begin{array}{c} 0\\ 0\\ 0\\ 0\\ 0\\ 0\\ 0\\ 0\\ p_{3,2}\\ U_{3,2} \end{array}\right].$$

The transfer matrix for section (0_2) can be written as follows:(40)$$\left[\begin{array}{c} p_{1,0}\\ U_{1,0} \end{array}\right]=\left[\begin{array}{cc} R_{1,9} & R_{1,10}\\ R_{2,9} & R_{2,10} \end{array}\right]\left[\begin{array}{c} p_{3,2}\\ U_{3,2} \end{array}\right],$$
(41)$$\left[\begin{array}{c} p_{2,0}\\ U_{2,0} \end{array}\right]=\left[\begin{array}{cc} R_{3,9} & R_{3,10}\\ R_{4,9} & R_{4,10} \end{array}\right]\left[\begin{array}{c} p_{3,2}\\ U_{3,2} \end{array}\right].$$

Section (2_3) contains a cavity; thus, its transfer matrix is expressed as follows:(42)$$\left[\begin{array}{c} p_{3,2}\\ U_{3,2} \end{array}\right]=\left[\begin{array}{cc} \cos\left(\bar{k}L_{c}\right) & i\bar{Z}\sin\left(\bar{k}L_{c}\right)\\ \frac{i\sin\left(\bar{k}L_{c}\right)}{\bar{Z}} & \cos\left(\bar{k}L_{c}\right) \end{array}\right]\left[\begin{array}{c} p_{4,3}\\ U_{4,3} \end{array}\right].$$

Section (3_4) is composed of homogeneous elastic materials, and its transfer matrix is $T_{5,3\_4}$. Finally, the transfer matrices of all sections are multiplied as follows to obtain the relationships between the acoustic variables of the incident and terminal surfaces of the overall structure:(43)$$\left[\begin{array}{c} p_{1,0}\\ U_{1,0} \end{array}\right]=\left[T_{1,0\_1}\right]\left[T_{3,1\_2}\right]\left[T_{4,2\_3}\right]\left[T_{5,3\_4}\right]\left[\begin{array}{c} p_{5,4}\\ U_{5,4} \end{array}\right],$$
(44)$$\left[\begin{array}{c} p_{2,0}\\ U_{2,0} \end{array}\right]=\left[T_{2,0\_1}\right]\left[T_{3,1\_2}\right]\left[T_{4,2\_3}\right]\left[T_{5,3\_4}\right]\left[\begin{array}{c} p_{5,4}\\ U_{5,4} \end{array}\right].$$

Under the assumption of a rigid backplane, the volume velocity at the bottom of the structure is 0 (i.e., $U_{5,4}=0$). The specific acoustic impedance of the overall system can be obtained using the parallel impedance rule as follows:(45)ZT=S3,1_2(S1,0_1(p1,0/U1,0))+(S2,0_1(p2,0/U2,0)).

The reflection coefficient at the incident surface of the structure can be determined using the following equation:(46)$$R=\frac{Z_{T}-\rho_{w}c_{w}}{Z_{T}+\rho_{w}c_{w}},$$
where $\rho_{w}$ is the density of water and $c_{w}$ is the sound velocity in water. The absorption coefficient α of the overall structure can be expressed as follows:(47)$$\alpha =1-{\left|R\right|}^{2}.$$

## 3. Results and Discussion

### 3.1. Numerical Calculations

To verify the theoretical results, we used COMSOL Multiphysics to conduct finite-element simulations. The three-dimensional model used for the finite-element simulations is displayed in Figure 5a. The acoustic–solid coupling module of the aforementioned software was adopted to estimate the sound absorption coefficient of the developed hybrid sound-absorbing metastructure. The elastic material domain was simulated using the elastic waves component, and the air in the internal cavity and the external water were simulated using the pressure acoustics component. The average sound pressure and axial velocity in plane A were evaluated to calculate the sound absorption coefficient. However, because of the thermal viscous dissipation of the MPP, the thickness of the boundary layer had to be considered in the finite-element software, which resulted in a high calculation cost. Therefore, internal perforated conditions were used in this study to model the MPP, and the fluid in the MPP was set as water. Rigid boundaries were set at the bottom of the metastructure and the outer boundary of the water-filled cavity. Moreover, the radial displacement field of the outer boundary of the elastomer was defined as a fixed constraint. Plane sound waves were incident from above the hybrid metastructure, as depicted in Figure 5b.

Typical viscoelastic polymer materials such as rubber and polyurethane are commonly used as substrates. The characteristic impedances of these substrates are close to that of water; thus, they can effectively reduce the reflection of sound waves. Moreover, the aforementioned materials have suitable sound absorption properties because of their viscosity and elasticity. Many elastomers with strong sound absorption effects have been proposed. In this study, the material parameters in [8] were considered as the parameters of the substrate of the developed sound-absorbing metastructure (Table 1). The detailed geometric parameters of the hybrid sound-absorbing metastructure are shown in Figure 6 and presented in Table 2.

### 3.2. Verification

Before employing the comparison of the theoretical and simulated absorption coefficients, it is necessary to understand the influence of metastructure without MPP on the sound absorption. Figure 7 displays the absorption coefficient profile of the anechoic structure with cavities. A sound-absorbing peak appears because of the resonant effect. The absorption coefficient peak of the typical Alberich anechoic coating structure with cylinder cavity [22] appears at 5.2 kHz in the simulations. The anechoic structure has an improved absorption performance at the low-frequency region when combined with the microporous plate, and the original sound absorption peak moves to the high-frequency region. Moreover, the sound absorption coefficient of the hybrid anechoic structure can reach more than 0.8 in the broadband range applied by sonar, which has a fairly good sound absorption effect.

Figure 8 displays the influence of the cavity thickness on sound absorption. As the cavity thickness $L_{c}$ increased from 7.5 to 12.5 mm, the theoretical results did not contain an absorption peak in the frequency band of 2–4 kHz. In the high-frequency region, as the cavity thickness increased, the total stiffness of the overall sound-absorbing metastructure decreased; thus, the second sound absorption peak moved toward the low-frequency region, whereas the first peak still exhibited the same resonance frequency. The results indicated that the developed metastructure has different sound absorption mechanisms. Figure 9 displays the influence of the cavity thickness to the sound absorption against the excited sonar frequency. As the cavity thickness $L_{c}$ increased from 0.005 to 0.04 m at different frequencies, a broadband absorption interval at 0.01 m could be observed.

Figure 10 displays the theoretical predictions and finite-element simulations obtained under different cavity radii ($r_{c}$). For $r_{c}$ = 10 mm, the theoretical sound absorption coefficient curve in Figure 10 does not include the first resonance peak of the simulated (FEM) sound absorption coefficient curve. When the cavity radius $r_{c}$ was 1 mm, the peak of the sound absorption coefficient appeared in the high-frequency region. By contrast, when the radius of the cavity $r_{c}$ was 20 mm, the stiffness of the overall metastructure decreased; thus, the absorption peak shifted toward low frequencies. Figure 11 displays the influence of cavity diameters on the contour profile of sound absorption. As the cavity diameters increased from 0.01 to 0.05 m at different frequencies, the peak absorption gradually moved towards the low-frequency region.

Figure 8 and Figure 10 indicate that the designed metastructure with MPP sound absorption peak occurred approximately at 3.1 kHz. Note that the absorption coefficient peak of the anechoic structure with cavities without MPP appears at 5.2 kHz (Figure 7). The preliminary results show that the MPP could be useful for shifting the absorption peak to the low-frequency region with more than 2 kHz frequency band. Overall, the theoretical and simulation results exhibit different peak frequencies in the low-frequency range of 1–3 kHz; however, the theoretical and simulation results for the middle- and high-frequency ranges (3–10 kHz) are similar. The theoretical results for the sound absorption coefficients are conservative. In addition, the difference between the theoretical and simulation results was mainly caused by the simplified analytical model adopted in this study. Nevertheless, the peak mismatch between the simulated and theoretical curves is acceptable.

### 3.3. Sound Absorption Mechanisms

This section describes the sound absorption mechanisms of the developed hybrid sound-absorbing metastructure. As displayed in Figure 12, when the cavity radius was 10 mm, absorption coefficient peaks occurred at 3.1 and 5.5 kHz in the simulations. Figure 12a,b display the acoustic intensity distributions at 3.1 and 5.5 kHz, respectively. At 3.1 kHz, sound energy was attracted to the MPP because the fluid in the cavities of the MPP caused a local resonance effect; thus, sound energy was dissipated at 3.1 kHz. At 5.5 kHz, no acoustic energy attraction was observed, and most of the sound energy was absorbed by the substrate of the hybrid metastructure.

Figure 13a shows the axial and radial displacement distributions for the hybrid metastructure at 3.1 kHz. According to this figure, most of the acoustic energy was absorbed by the MPP to achieve local resonance at 3.1 kHz. Therefore, a large axial displacement variation occurred under the panel, whereas the radial displacement distribution at the low-frequency peak did not exhibit considerable displacement. At 5.5 kHz [Figure 13b], no high-order mode was generated in the axial displacement and radial displacement distributions. Large radial displacement only occurred at the air-filled cavity wall. Because the radial displacement represents the vibration amplitude of the transverse wave, the existence of a cavity causes the generation of transverse waves at the resonance frequency of the overall structure. When the overall structure resonates, the cavity wall exhibits shear deformation. At this time, longitudinal waves are converted into transverse waves, which causes the sound energy to be quickly dissipated into the medium. Therefore, the existence of a cavity increases the sound absorption in the high-frequency region.

### 3.4. Numerical Parameter Studies

Studies [15,16,17] have indicated that the sound absorption performance of a sound-absorbing metastructure mainly depends on the material parameters of the structure. Figure 14 displays the influence of the real part of the Young’s modulus of the substrate on the acoustic performance of the developed hybrid metastructure. When the real part of Young’s modulus was increased, the sound absorption coefficient decreased marginally in the frequency range of 1–5 kHz but increased gradually in the frequency range of 5–10 kHz. As the real part of Young’s modulus was increased, the stiffness of the overall structure also increased, which resulted in the vibration mode of the structure moving to higher frequencies. Figure 15 displays the influence of the real part of Young’s modulus on contour of sound absorption. When the real part of Young’s modulus was 400 MPa, a strong high-frequency absorption band appeared.

In contrast to the Young’s modulus of elastic materials, the Young’s modulus of viscoelastic materials has an imaginary part. The mechanical energy generated by sound wave vibration can convert energy into heat in a viscoelastic material, which increases the absorption of sound by the material. The imaginary part of Young’s modulus is also called the loss factor, which can be regarded as the damping of the material. Figure 16 depicts the influence of the loss factor on the hybrid metastructure’s absorption coefficient. The first peak of the sound absorption coefficient was almost unchanged under different loss factor values. In the frequency range of 3–10 kHz, when the loss coefficient increased, the sound absorption coefficient also increased.

Figure 17 illustrates the influence of the density of the substrate on the hybrid metastructure’s sound absorption coefficient. Because the sound absorption peak in the low-frequency region was dominated by the MPP, the sound absorption coefficient in the high-frequency region was affected by the cavities. The sound absorption coefficient curve changed marginally at different substrate densities in the frequency range of 1–3 kHz. As the density of the substrate increased, the higher mass of the overall structure shifted the vibration mode to the low-frequency region; thus, the three curves in Figure 13 show the peak shift fluctuation that occurred in the frequency range of 3–10 kHz.

Compared with the traditional muffler, the MPP structure has a wider sound-absorbing belt and is often used for low-frequency sound energy absorption. However, for the microporous plate applied underwater, the perforation diameter of the corresponding micro-perforated plate is extremely small due to the small thickness of the fluid viscosity boundary layer. Considering the manufacturing process, we choose the perforation diameter and the plate thickness as 0.8 mm and 3 mm, respectively, and calculate the sound absorption profile of the hybrid metastructure. Figure 18 shows that the first absorption peak moves to the high-frequency region with the increase in porosity, with better absorption effect. The second absorption peak mostly remains unchanged. Nevertheless, the practical fabrication must ensure that the distance between the holes Is greater than its own hole diameter to prevent the interaction between the holes, so the porosity is suggested to be 0.0875, in this particular case, to accommodate a broadband and favorable absorption effect.

The thickness of micro perforated plate $t_{m}$ is proportional to the specific acoustic impedance provided by the microporous plate. Figure 19 illustrates the influence of the microporous plate thickness on the sound absorption coefficient of the proposed hybrid metastructure. When the microporous plate thickness changes, the change mainly affects the first peak of the absorption coefficient, but the coefficient profile is between 5 and 10 kHz. Considering the broadband absorption effect (e.g., absorption coefficient above 0.8), we set the thickness of the microporous plate to 3 mm.

### 3.5. Practical Underwater Applications

In the simulations of the design of the MPP, we assumed that internal perforated plate conditions existed in the sound pressure module. Therefore, the designed sound-absorbing metastructure must contain a water-filled cavity behind the MPP. In practical applications, the existence of a water-filled cavity tends to reduce overall structural strength. Therefore, we replaced the water-filled cavity in the original hybrid sound-absorbing metastructure with a homogeneous elastomer and examined the modified metastructure’s sound absorption coefficient. In this structure, only the holes of the MPP still contained fluid. Figure 20a depicts the sound absorption coefficients of the original and modified hybrid sound-absorbing metastructures. The impedance of water and that of the viscoelastic material are close; thus, the two sound absorption coefficient curves are consistent. To prevent the holes of the MPP from being damaged by seawater erosion under complex sea conditions, a layer of the adopted homogeneous elastomer was used to cover the top of the modified hybrid metastructure to increase the structure’s compression resistance and corrosion resistance. Figure 20b illustrates the sound absorption results obtained when adding a 5 mm thick elastomer in front of the modified hybrid metastructure. This elastomer had the sound absorption effect of a homogeneous sound-absorbing metastructure. Thus, the sound absorption performance of the modified hybrid metastructure was improved over a wide frequency range. Figure 21 illustrates the practical applications of the structural combination process.

## 4. Conclusions

In general, viscoelastic materials reduce the reflection of sound waves underwater when these materials exhibit good impedance matching with sea water, and friction is generated when sound waves enter these materials, which results in sound energy absorption. However, designing an underwater sound absorber with broadband sound absorption ability is challenging.

In this study, the transfer matrix method was used for rapidly evaluating a designed hybrid sound-absorbing metastructure. An MPP was introduced into this structure to solve the problem of insufficient low-frequency sound absorption. The metastructure designed in this study can achieve sound absorption over a wider bandwidth than can sound-absorbing metastructures described in the literature. The accuracy of the theoretical and simulation results was verified by comparing them with each other. The theoretical and simulation results exhibit different peak frequencies in the low-frequency range of 1–3 kHz; however, the theoretical and simulation results for the middle- and high-frequency ranges (3–10 kHz) are similar. The theoretical results for the sound absorption coefficients are conservative. In addition, the difference between the theoretical and simulation results was mainly caused by the simplified analytical model adopted in this study. Nevertheless, the peak mismatch between the simulated and theoretical curves is acceptable.

Moreover, a method for the practical application of the designed metastructure is described in this article. According to the results of this study, the designed metastructure has potential for use as an underwater sound-absorbing metastructure. Because of the high hydraulic pressure in the deep sea, the cavity of the designed metastructure is easily compressed, and its absorption peaks tend to occur in a high-frequency range. In a follow-up study, we will add a honeycomb support structure to the outer boundary of the designed hybrid structure to improve the design of the hybrid metastructure to enable its use at large water depths.

## Figures and Tables

**Figure 1 materials-16-01718-f001:**
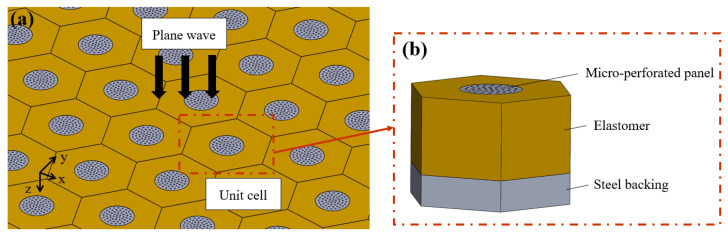
Schematic of developed hybrid sound-absorbing metastructure: (**a**) configuration and problem description and (**b**) unit cell of the metastructure.

**Figure 2 materials-16-01718-f002:**
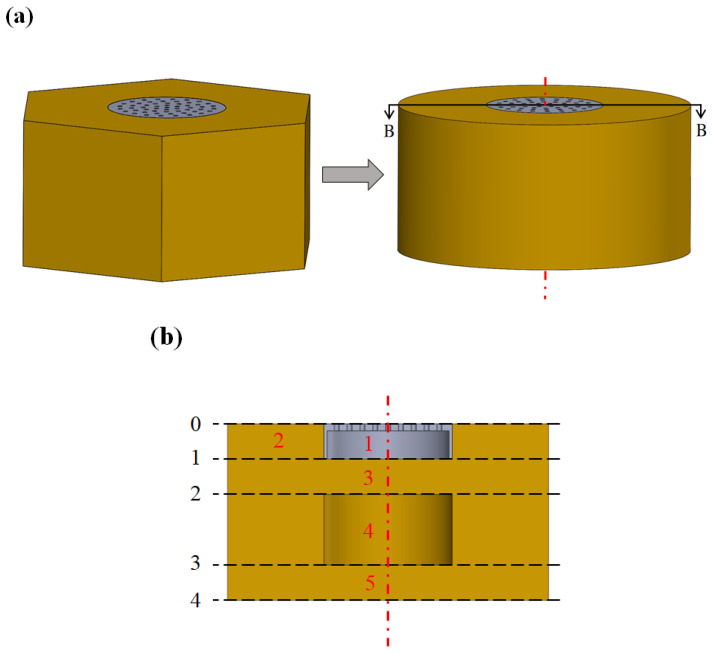
(**a**) Schematic of approximate model of developed hybrid sound-absorbing metastructure and (**b**) two-dimensional plot of cut plane B–B of multiple sections of hybrid sound-absorbing structure.

**Figure 3 materials-16-01718-f003:**
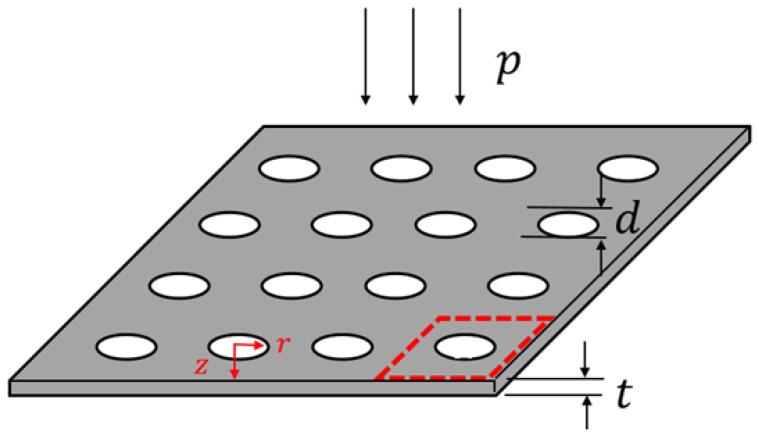
Schematic of the microperforated panel (MPP).

**Figure 4 materials-16-01718-f004:**
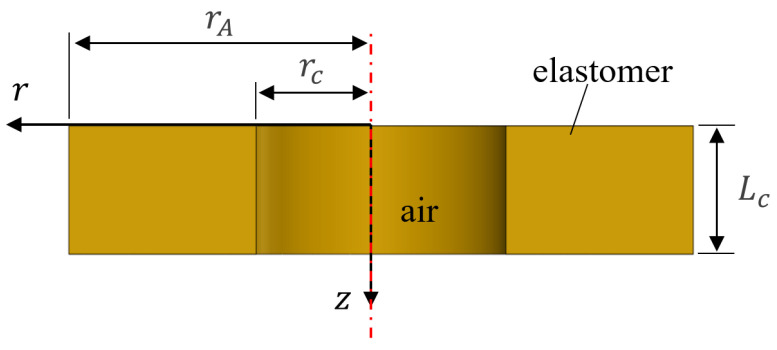
Profile of a section with an air-filled cavity.

**Figure 5 materials-16-01718-f005:**
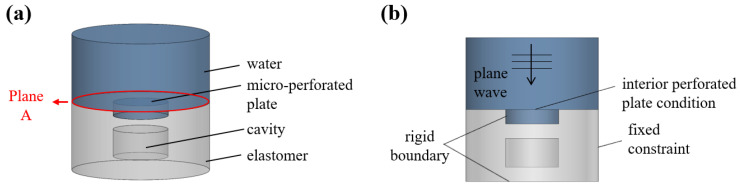
Simulation model for analyzing the sound absorption coefficient of the developed hybrid sound-absorbing metastructure: (**a**) simulation setup and (**b**) boundary conditions.

**Figure 6 materials-16-01718-f006:**
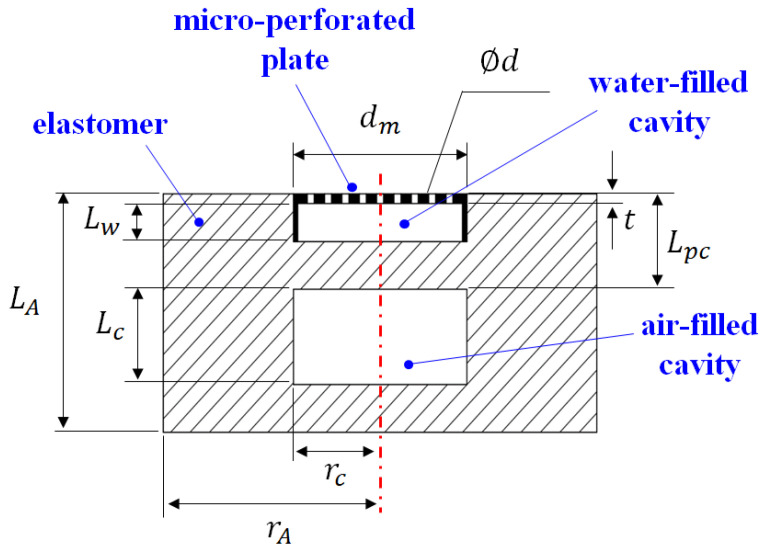
Schematic and geometric parameters of the hybrid sound-absorbing metastructure in finite-element simulations.

**Figure 7 materials-16-01718-f007:**
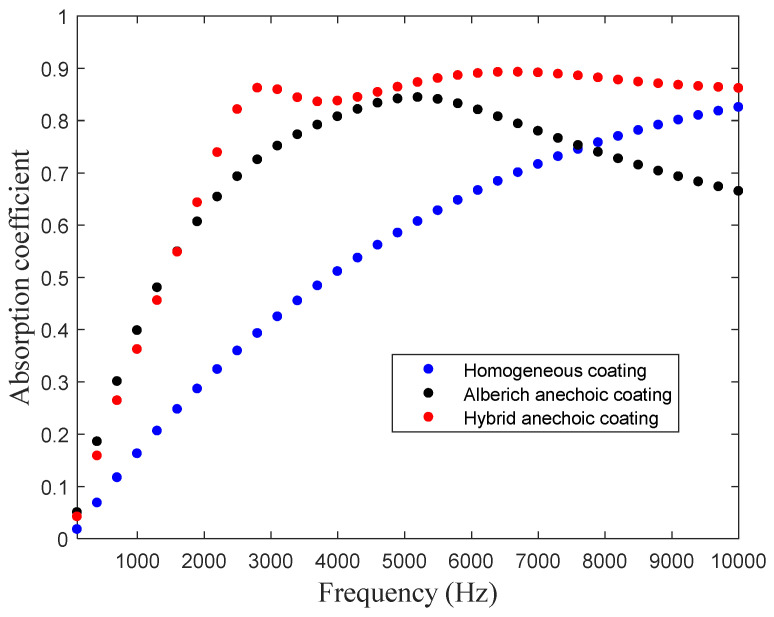
Influence of metastructure without MPP on sound absorption.

**Figure 8 materials-16-01718-f008:**
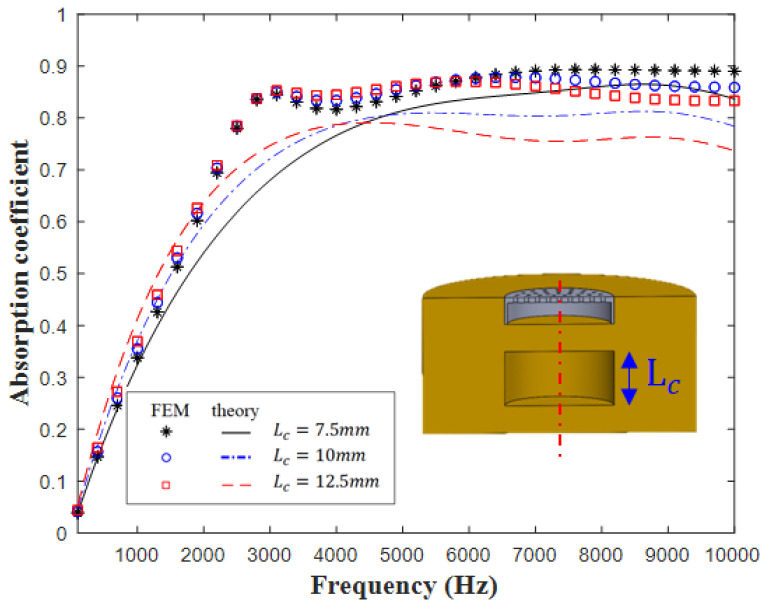
Influence of cavity thickness on sound absorption.

**Figure 9 materials-16-01718-f009:**
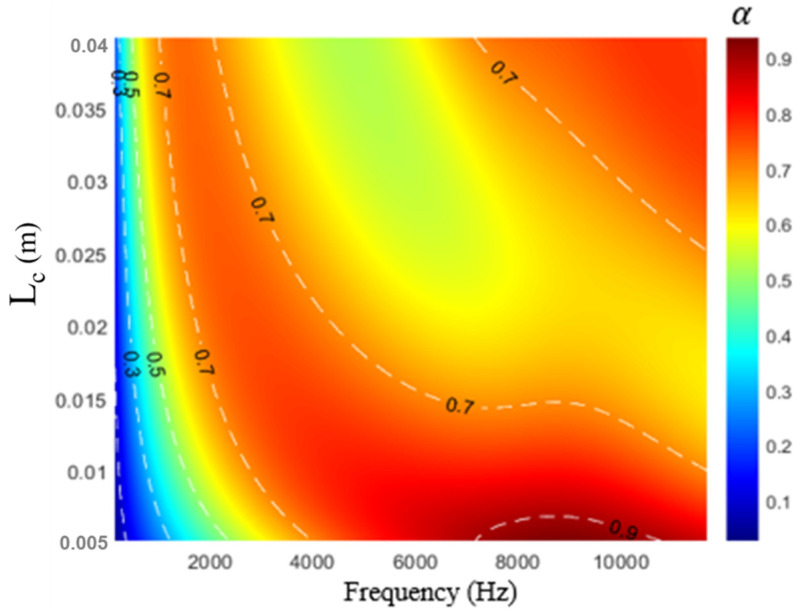
Influence of cavity thickness on contour of sound absorption.

**Figure 10 materials-16-01718-f010:**
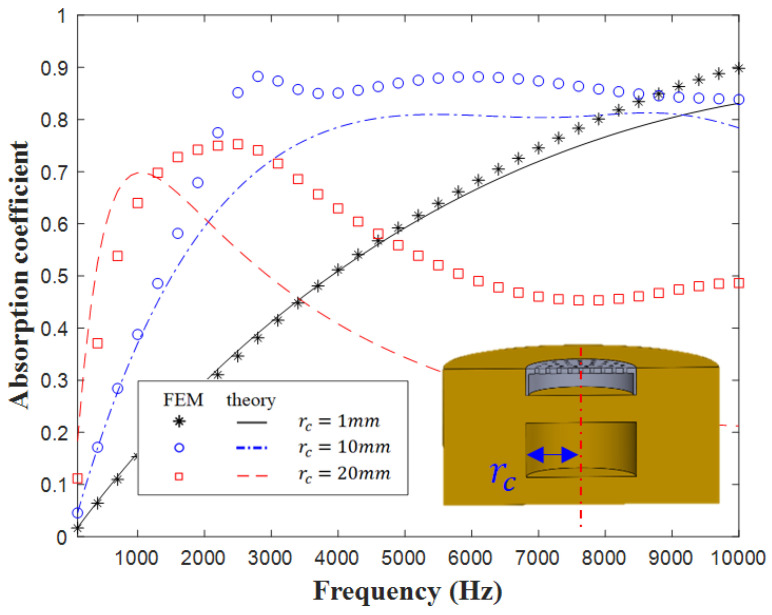
Influence of cavity radius on sound absorption.

**Figure 11 materials-16-01718-f011:**
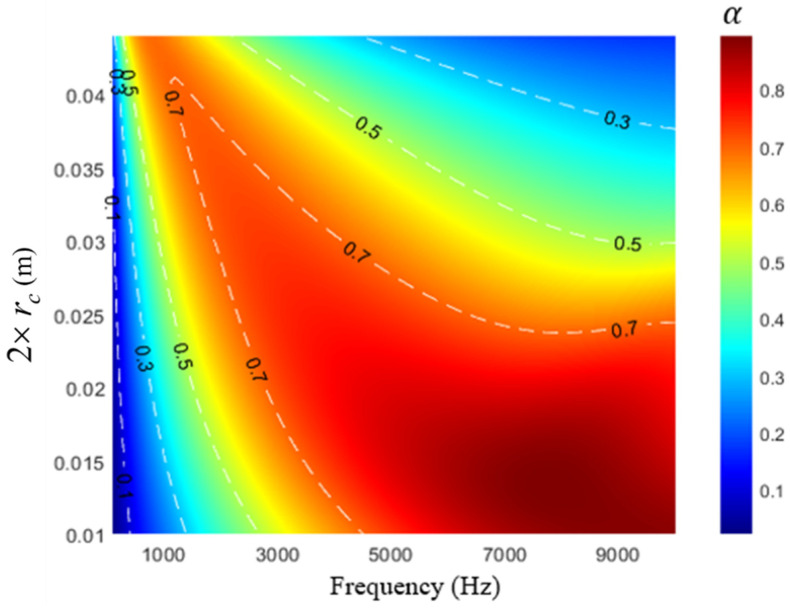
Influence of cavity diameters on contour sound absorption.

**Figure 12 materials-16-01718-f012:**
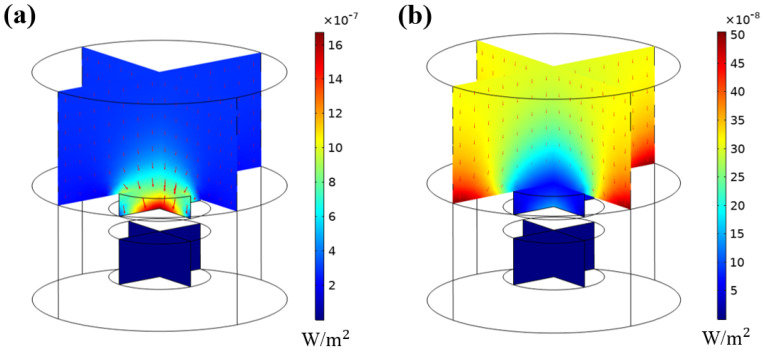
Acoustic intensity distributions for the hybrid sound-absorbing metastructure at (**a**) 3.1 and (**b**) 5.5 kHz.

**Figure 13 materials-16-01718-f013:**
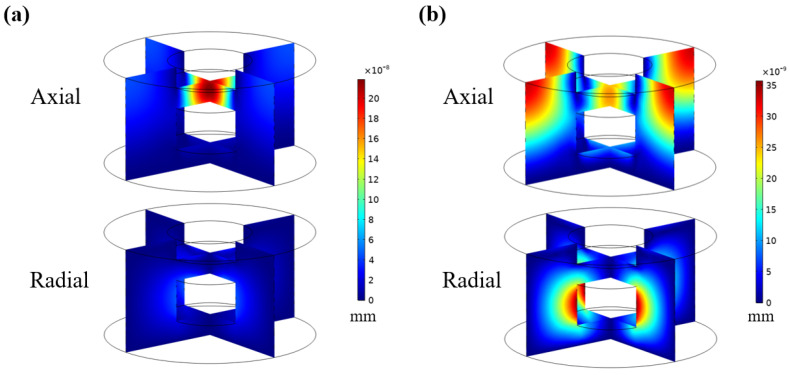
Axial and radial displacement distributions for the hybrid sound-absorbing metastructure at (**a**) 3.1 and (**b**) 5.5 kHz.

**Figure 14 materials-16-01718-f014:**
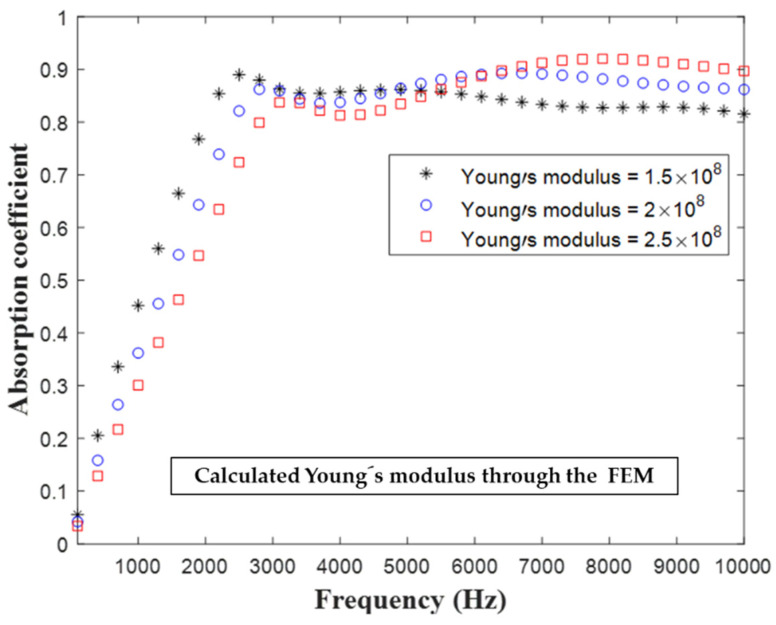
Influence of the real part of Young’s modulus (Unit: Pa) of the substrate of the developed metastructure on the structure’s sound absorption coefficient.

**Figure 15 materials-16-01718-f015:**
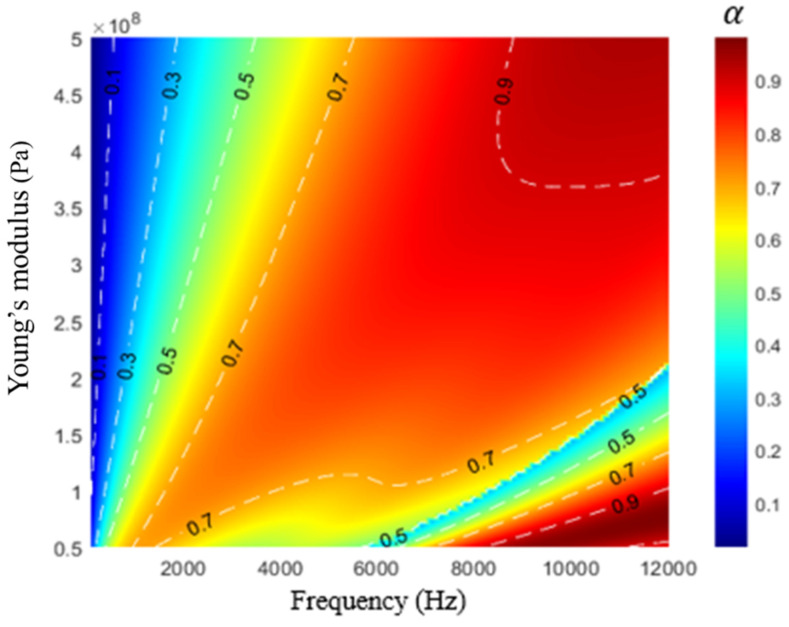
Influence of the real part of Young’s modulus on sound absorption.

**Figure 16 materials-16-01718-f016:**
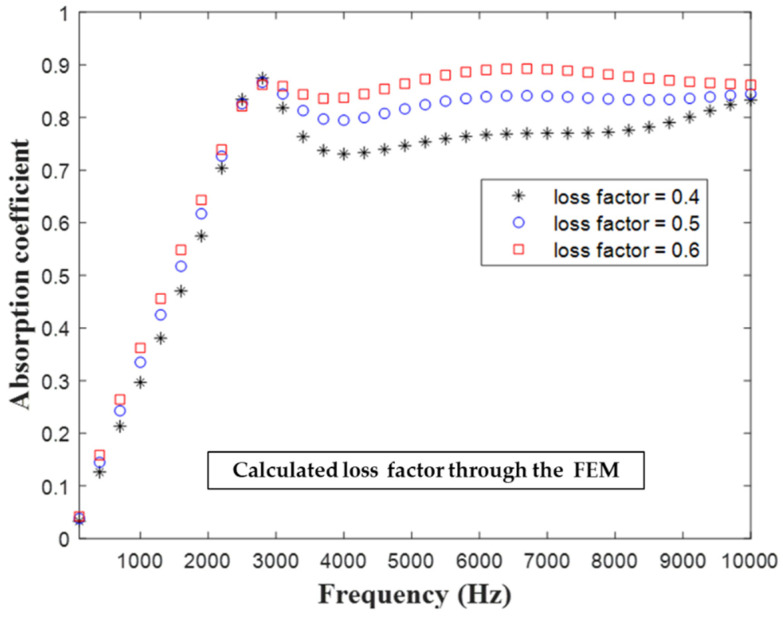
Influence of the loss factor of the substrate of the hybrid metastructure on the structure’s sound absorption coefficient.

**Figure 17 materials-16-01718-f017:**
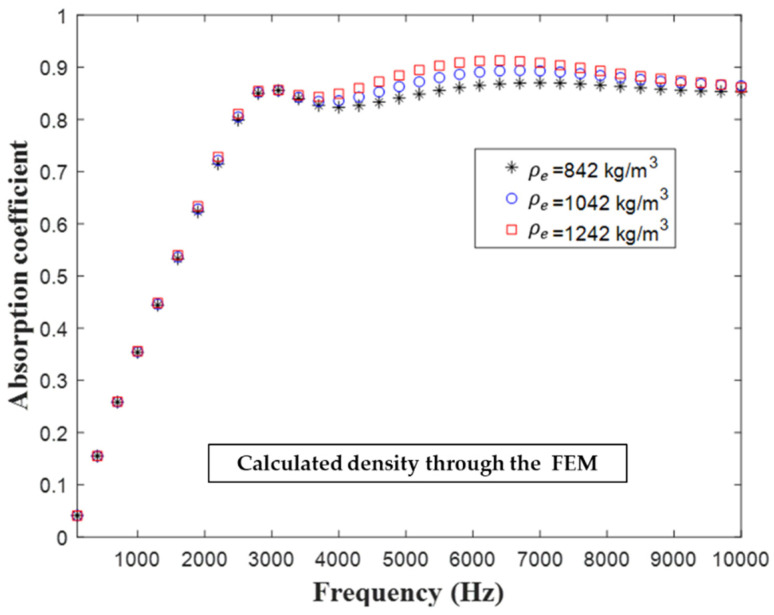
Influence of substrate density of the substrate of the hybrid metastructure on the structure’s sound absorption coefficient.

**Figure 18 materials-16-01718-f018:**
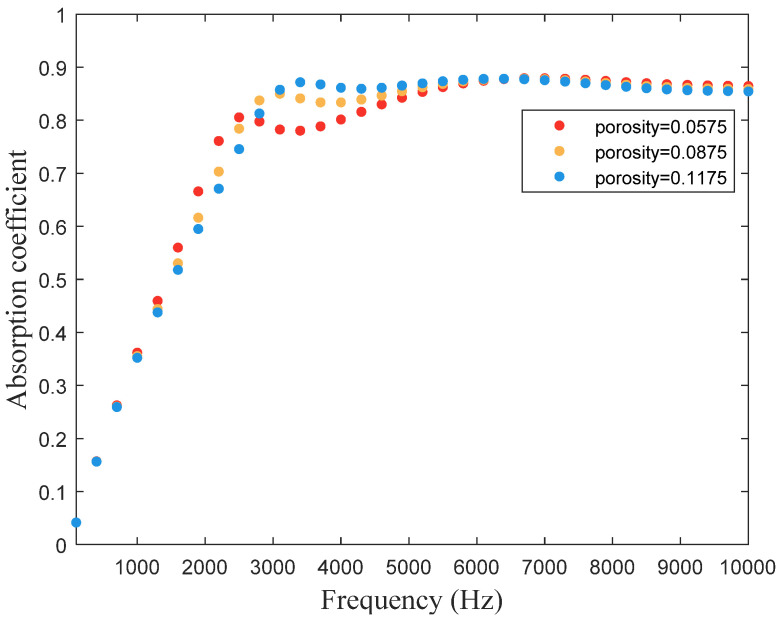
Influence of porosity of micro perforated plate of the hybrid metastructure on the structure’s sound absorption coefficient.

**Figure 19 materials-16-01718-f019:**
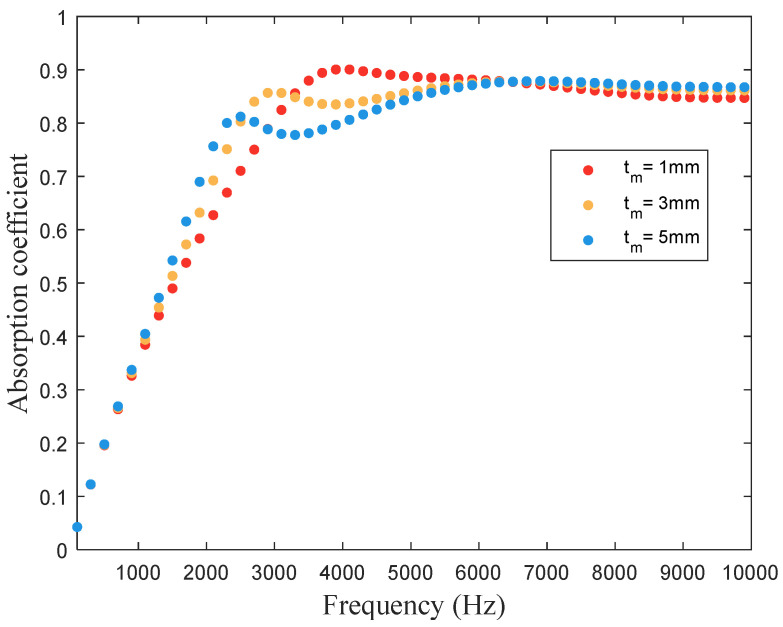
Influence of thickness of the micro perforated plate of the hybrid metastructure on the structure’s sound absorption coefficient.

**Figure 20 materials-16-01718-f020:**
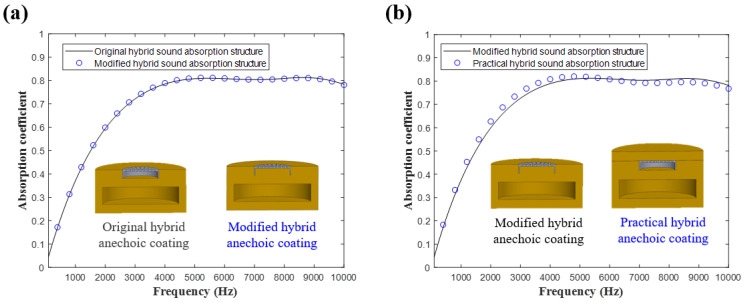
Sound absorption coefficients of (**a**) original and modified hybrid sound-absorbing metastructures and (**b**) modified and practical hybrid sound-absorbing metastructures.

**Figure 21 materials-16-01718-f021:**
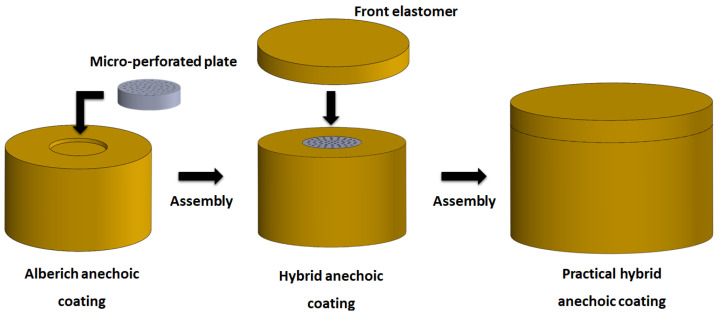
The practical applications of metastructure assembly process.

**Table 1 materials-16-01718-t001:** Parameters considered for the substrate of the developed metastructure [8].

Quantity	Values
Density of the elastomer ($\mathrm{kg}/m^{3}$)	1042
Young’s modulus of the elastomer (Pa)	$2\times 10^{8}$
Loss factor of the elastomer	0.6
Poisson ratio of the elastomer	0.485
Density of air ($\mathrm{kg}/m^{3}$)	1.2
Density of water ($\mathrm{kg}/m^{3}$)	998
Sound speed in water (m/s)	1481

**Table 2 materials-16-01718-t002:** Geometric parameters of the hybrid sound-absorbing metastructure in finite-element simulations.

Parameter	Values
$L_{u}$ (mm)	150
$d_{u}$ (mm)	50
$L_{A}$ (mm)	25
$r_{A}$ (mm)	25
$L_{c}$ (mm)	10
$L_{w}$ (mm)	5
$r_{c}$ (mm)	10
$L_{pc}$ (mm)	10
d (mm)	0.8
ϕd	0.0875
t (mm)	3
$d_{m}$ (mm)	20

## Data Availability

The data presented in this study are available on request from the corresponding author.

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
