# Peer review of "Evaluation of a Hybrid Underwater Sound-Absorbing Metastructure by Using the Transfer Matrix Method"

_materials, 2023, doi:10.3390/ma16041718_

Round 1
Reviewer 1 Report
The article is well written and addresses a current issue.
However, the insertion of MPP which is supposed to improve the performance at low frequencies is not clearly demonstrated. - What is the proportion of dissipated energy by MPP and its influence in the curves of Figure 7 and Figure 8? - In the following, it is shown that the MPP is active at 3.1 kHz but its effect seems limited and is not quantified with respect to the attenuation obtained by the viscoelastic layer. - Finally, section 3.5 does not clearly show the effect of the MPP.
Moreover, the conclusion does not meet the title: the article does not make an evaluation of the composite structure by the TMM method but rather by FEM with Comsol.
For this reason, I do not recommand to publish this paper in its present form.
Detailed comment:
1. In the abstract, the sentence "The acoustic performance ... sound-absorbing structure" is too long.
2. Assumptions of the TMM method should be given when dealing with concentric (or paralell) media, like in sections 0_1 and 2_3. Is the proposed method identical to the so-called PTMM (see reference below)?
Reference: Verdière, K., Panneton, R., Elkoun, S., Dupont, T. & Leclaire, P. Transfer matrix method applied to the parallel assembly of sound absorbing materials. The Journal of the Acoustical Society of America 134, 4648-4658 (2013).
Reviewer 2 Report
In general, the subject treated in the article "Evaluation of a Hybrid Underwater Sound-Absorbing Metastructure by Using the Transfer Matrix Method" by H.-C- Lin, et al. is very interesting. The article shows novel results in the field of sound-absorbing metastructures. However, the article needs minor changes to be published in Materials.
The following is a list of observations:
1. Abstract: write the materials of the metastructure, which are shown in Figure 1.
2. Keywords: add the terms “microperfored panel” and “viscoelastic substrate”.
3. The information in the inner box of the Figure 11 is confused; it is recommendable to write the title as “Calculated Young´s Modulus”, below this write FEM. Same situation for the inner box in Figure 12, for loss factor, and in figure 13 for the density.
Reviewer 3 Report
A metastructure comprising cells of perforated panels surrounded by elastometers is proposed to absorb sound underwater. With use of transfer matrix method the absorption coefficient is computed and simulated with good agreement between the two sets of data. The structural parameters of the design are determined and various radii and heights of the cavity are considered. The displacement distributions demonstrate the absorption for several oscillating frequencies.
The paper is extensive and elaborates a crucial issue like the underwater sound absorption. However, certain weak points should be fixed prior recommending publication at MDPI Materials. More specifically:
(A) The authors should thoroughly explain how all these parameters are determined. Is there something optimal in this selection?
(B) Contour plots of the absorbance with respect to the most crucial design parameters at a fixed frequency should be provided and commented.
(C) Can the authors achieve similar results without patterned surface but with simpler homogeneous layers with curved [1] or planar [2] shape? Comparison graphs on why perforated designs are preferable should be included in the revised version.
(D) The authors may discuss the differences and the similarities between their structures and respective cloaking setups for electromagnetic [3] and acoustic [4] waves.
[1] Shell-type acoustic metasurface and arc-shape carpet cloak, Scientific Reports, 2019.
[2] Wide-angle absorption of visible light from simple bilayers, Applied Optics, 2017.
[3] Electromagnetic cloaking of cylindrical objects by multilayer or uniform dielectric claddings, Phys. Rev. B, 2012.
[4] Deterministic and probabilistic deep learning models for inverse design of broadband acoustic cloak, Phys. Rev. Res., 2021.
Round 2
Reviewer 1 Report
The reviewer thanks the authors for improvement of the paper. I appreciate Figure 7 but I still wonder what is the effect of the MPP in Figure 8. What happens without the MPP in those configurations? What dissipation ratio is due to the MPP itself? A figure with MPP having different holes radius d could answer to these questions, for instance. By the way, what is the holes radius used for simulations?
The paper may be published after answering these questions.
Minor comments:
1. In figure 7, "Alberich" term is used: to be defined in the paper.
2. Line 287, 302: add a space between number and unit "0.04 mm".
Author Response
The response letter is attached.
